# Sequentially induced motor neurons from human fibroblasts facilitate locomotor recovery in a rodent spinal cord injury model

Hyunah Lee[1,2], Hye Yeong Lee[3], Byeong Eun Lee[2], Daniela Gerovska[4], Soo Yong Park[1,2], Holm Zaehres[5], Marcos J Araúzo-Bravo[4,5,6], Jae-Ick Kim[2], Yoon Ha[3], Hans R Schöler[5], Jeong Beom Kim[1,2]*

[1]Hans Schöler Stem Cell Research Center (HSSCRC), Ulsan National Institute of Science and Technology (UNIST), Ulsan, Republic of Korea; [2]School of Life Sciences, Ulsan National Institute of Science and Technology (UNIST), Ulsan, Republic of Korea; [3]Department of Neurosurgery, Spine and Spinal Cord Institute, Severance Hospital, Yonsei University College of Medicine, Seoul, Republic of Korea; [4]Computational Biology and Systems Biomedicine Group, Computational Biomedicine Data Analysis Platform, Biodonostia Health Research Institute, San Sebastián, Spain; [5]Department of Cell and Developmental Biology, Max Planck Institute for Molecular Biomedicine, Münster, Germany; [6]IKERBASQUE, Basque Foundation for Science, Bilbao, Spain

**Abstract** Generation of autologous human motor neurons holds great promise for cell replacement therapy to treat spinal cord injury (SCI). Direct conversion allows generation of target cells from somatic cells, however, current protocols are not practicable for therapeutic purposes since converted cells are post-mitotic that are not scalable. Therefore, therapeutic effects of directly converted neurons have not been elucidated yet. Here, we show that human fibroblasts can be converted into induced motor neurons (iMNs) by sequentially inducing *POU5F1(OCT4)* and *LHX3*. Our strategy enables scalable production of pure iMNs because of the transient acquisition of proliferative iMN-intermediate cell stage which is distinct from neural progenitors. iMNs exhibited hallmarks of spinal motor neurons including transcriptional profiles, electrophysiological property, synaptic activity, and neuromuscular junction formation. Remarkably, transplantation of iMNs showed therapeutic effects, promoting locomotor functional recovery in rodent SCI model. Together, our advanced strategy will provide tools to acquire sufficient human iMNs that may represent a promising cell source for personalized cell therapy.

*For correspondence:
jbkim@unist.ac.kr

Competing interests: The authors declare that no competing interests exist.

## Introduction

Spinal cord injury (SCI) causes devastating neurological impairments and disabilities (*Singh et al., 2014*). SCI leads to the loss of sensory/motor functions and malfunctions in other organs such as bladder, kidneys and bowel, provoking an enormous impact on physical, psychological and social behavior of SCI patients (*Bradbury and McMahon, 2006*; *Vismara et al., 2017*). Unfortunately, there is no fully restorative treatment for SCI yet (*Silva et al., 2014*). Medications or surgical decompression are the only options for SCI treatment, however, these approaches are controversial due to the severe side effects and limited clinical efficacy (*Cristante et al., 2012*; *Vismara et al., 2017*).

Thus, cell replacement therapy has been proposed as a promising therapeutic intervention to reconstitute the damaged nervous system and improve functional recovery after SCI (*Goldman, 2005*). A number of studies have shown that transplantation of neural cells derived from pluripotent stem cells (PSCs) is effective in functional and histological restoration in SCI animal model (*Lu et al., 2014*; *Nakamura and Okano, 2013*; *Tsuji et al., 2010*; *Vismara et al., 2017*). Despite these encouraging advances, ethical issue of embryonic stem cells (ESCs) and tumorigenic potential of induced pluripotent stem cells (iPSCs) have impeded their translations into clinical trials (*Deng et al., 2018*; *Fong et al., 2010*; *Miura et al., 2009*; *Ronaghi et al., 2010*). To overcome these limitations, direct conversion of somatic cells into other cell types has been achieved while bypassing pluripotent state (*Park et al., 2019*; *Park et al., 2020*; *Xu et al., 2015*). However, application of previous protocols hindered therapeutic translations due to the heterogeneity and low yields, and involvement of multiple transcription factors which may increase the genetic mutagenesis (*Kang et al., 2015*). Most importantly, directly converted cells are fully differentiated post-mitotic cells which limit the acquisition of cell source in large-scale for therapeutic purpose. Hence, there is a critical need to develop a new method that enables the large-scale production of highly pure and functional target cells with consistent quality. In terms of treating SCI, it must be capable of generating motor neuron (MN) subtypes that are relevant to disease rather than producing general neurons.

In this study, we succeeded in generating induced motor neurons (iMNs) by using a minimal number of transcription factors, *POU5F1(OCT4)* and *LHX3*. *POU5F1(OCT4)* is known to play an important role in regulating pluripotent genes (*Shi and Jin, 2010*; *Wang et al., 2007*), and downstream target genes involved in developmental processes (*Shi and Jin, 2010*). Previously, overexpression of *POU5F1(OCT4)* allowed the generation of blood progenitor cells from fibroblasts (*Szabo et al., 2010*) by regulating hematopoietic gene, *HOXB4*, which is one of the *POU5F1(OCT4)* targets (*Boyer et al., 2005*). Furthermore, a number of studies showed that *POU5F1(OCT4)* can induce various cell fate reprogramming such as neural stem cells into iPSCs (*Kim et al., 2009a*; *Kim et al., 2009b*), and fibroblasts into neural progenitor cells (*Mitchell et al., 2014b*) as well as oligodendrocyte progenitor cells (*Kim et al., 2015*), defining *POU5F1(OCT4)* as a versatile reprogramming factor that confers the plasticity in somatic cells (*Mitchell et al., 2014a*; *Mitchell et al., 2014b*). Also, it has been reported that *POU5F1(OCT4)* binds to homeodomain transcription factor *ISL1* (*Boyer et al., 2005*; *Jung et al., 2010*) which is required for specification of spinal cord MNs (*Cho et al., 2014*; *Liang et al., 2011*). So, we hypothesized that activation of *POU5F1(OCT4)* might have potential to generate MNs from somatic cells through regulating *ISL1* expression. Here, we introduced the key cell fate regulator *POU5F1(OCT4)* and subsequently overexpressed additional MN specification factor *LHX3* to induce fibroblasts toward motor neuronal fate. Importantly, we found that iMNs exhibited typical characteristics of MNs on molecular level, electrophysiological activity, synaptic functionality, in vivo engraftment capacity and therapeutic effects. In conclusion, our strategy enables large-scale production of pure iMNs and facilitates the feasibility of iMNs for SCI treatment. Access to high-yield cultures of human MNs will facilitate an in-depth study of MN subtype-specific properties, disease modeling, and development of cell-based drug screening assays for MN disorders.

## Results

### Generation of induced motor neurons (iMNs) from human fibroblasts by sequential induction of two transcription factors

To generate induced motor neurons (iMNs) from human adult fibroblasts (HF1), we established an advanced direct conversion strategy by inducing *POU5F1(OCT4)* and *LHX3* sequentially at different time points as described in the experimental scheme (*Figure 1A*). To rule out the possibility that resulting iMNs are derived from contaminating neural cells in parental fibroblasts, we confirmed that none of the fibroblasts express neural lineage markers by immunostaining with specific antibodies (*Figure 1—figure supplement 1A*). Firstly, we transduced fibroblasts with *POU5F1(OCT4)* regarding its critical role in cell fate decision during early development (*Yamada et al., 2013*). *POU5F1(OCT4)*-induced plasticity has been shown to activate the lineage genes in response to lineage supporting culture condition (*Mitchell et al., 2014a*; *Mitchell et al., 2014b*). We cultured *POU5F1(OCT4)*-infected cells in our defined neural induction medium. The morphological changes were observed as

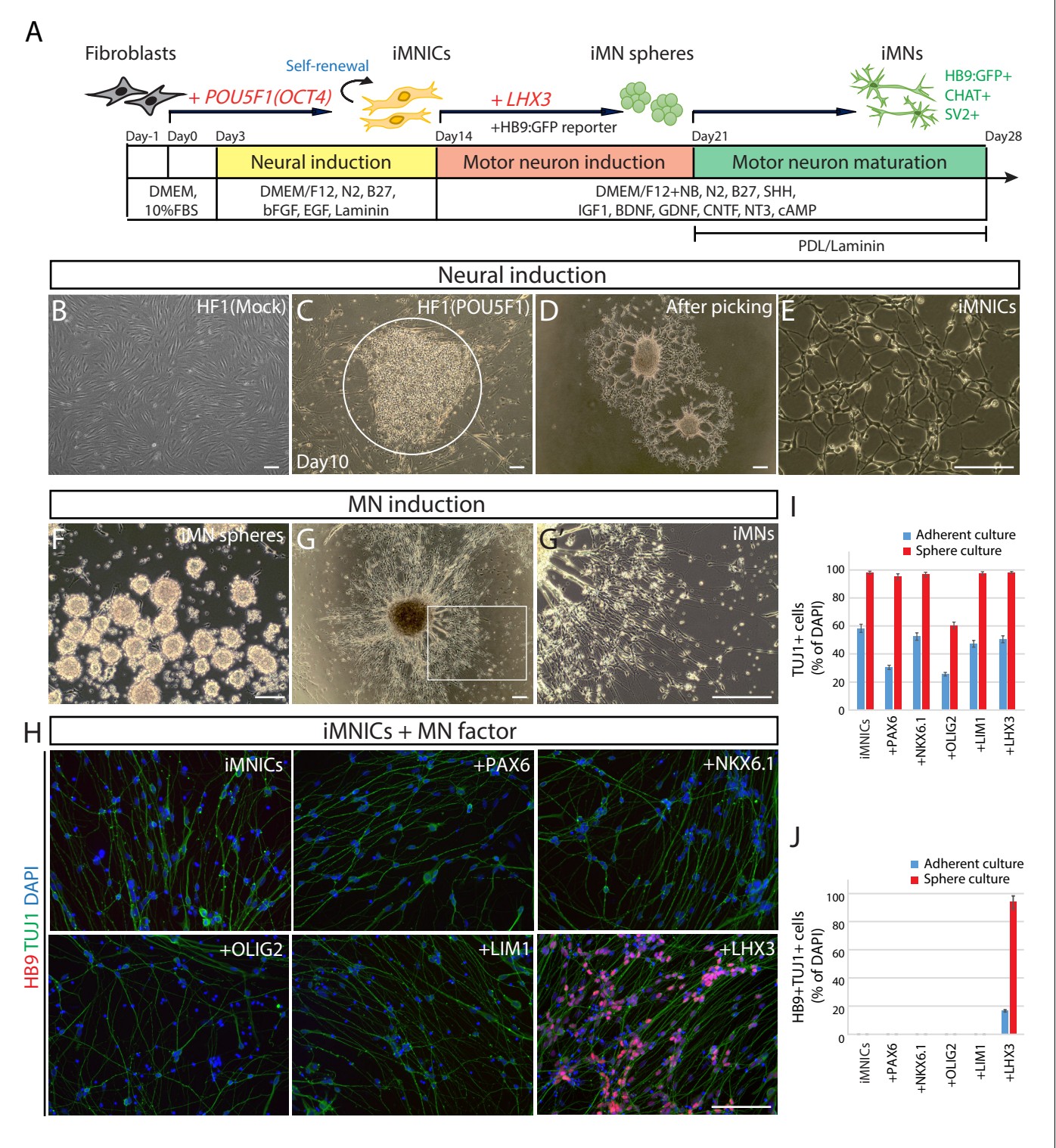

**Figure 1.** Generation of induced motor neurons (iMNs) from human fibroblasts. (**A**) The experimental scheme for the generation of iMNs by sequential transduction of two transcription factors. (**B–G**) The morphological change of human fibroblasts (HF1) during reprogramming. (**B**) The morphology of mock-infected HF1 in neural induction medium. (**C**) The emergence of cell cluster after 10 days of *POU5F1(OCT4)* infection in neural induction medium. (**D**) The morphology of *POU5F1(OCT4)*-infected cells after picking. (**E**) The morphology of iMNICs after passage. (**F**) Appearance of iMN spheres in motor neuron induction medium. (**G**) The morphology of iMNs on PDL/Laminin-coated plate. (**G′**) Zoomed image of the square in (**G**) shows complex axonal processes. Scale bars, 125 μm in (**B–G′**). (**H**) Immunofluorescence images of iMNs (HF1-iMN2) converted from iMNICs after 14 days of MN factor transduction under sphere culture condition. Addition of *LHX3* to iMNICs but not any other MN factors enhanced conversion efficiency to generate

*Figure 1 continued on next page*

*Figure 1 continued*

HB9+TUJ1+ iMNs. The cells were co-stained with HB9 (red) and TUJ1 (green), and the nuclei were counterstained with DAPI. Scale bars, 125 µm. (I–J) Conversion efficiency determined by quantification of TUJ1+ cells (I) and HB9+TUJ1+ cells (J) depending on transduced MN factor and culture condition. Data are presented as mean ± SD (n = 3), and represent triplicate experimental replicates. Related data can be found in *Figure 1—figure supplements 1–6*.

The online version of this article includes the following source data and figure supplement(s) for figure 1:

**Source data 1.** Quantification of TUJ1+/HB9+TUJ1+ iMNs depending on TF, qRT-PCR analysis of MN genes, and qRT-PCT analysis of endogenous expression of *POU5F1*, *NANOG*, and *SOX2*.

**Figure supplement 1.** Validation of parental human fibroblasts.

**Figure supplement 2.** Establishment of iMNIC clones and upregulation of *ISL1* expression after *POU5F1(OCT4)* induction.

**Figure supplement 3.** Characterization of iMNs converted by *POU5F1(OCT4)* alone.

**Figure supplement 4.** Screening transcription factor and optimization of culture condition for generating iMNs from iMNIC clones.

**Figure supplement 5.** iMNICs possess self-renewal capacity and do not transit neither pluripotent state nor neural progenitor state.

**Figure supplement 6.** Primary induction of *POU5F1(OCT4)* outperforms simultaneous induction of *POU5F1(OCT4)* and *LHX3*.

early as day 3, and spindle-shaped cells then formed compact colonies by day 10 (*Figure 1C*), whereas mock-infected fibroblasts did not change under the same condition (*Figure 1B*). From $0.5 \times 10^4$ seeded fibroblasts, we could observe 7 ~ 12 colonies (*Supplementary file 3*). We picked colonies individually and plated them in separate wells for clonal culture. The neural progenitor-like cells grew out gradually from attached colonies (*Figure 1D*). We found that the number of cells increased over time in neural induction medium. These cells were passaged and maintained as a monolayer cell line which we named iMN-intermediate cells (iMNICs) (*Figure 1E*). We could establish six iMNIC clones from HF1; HF1-iMNIC2, HF1-iMNIC5, HF1-iMNIC6, HF1-iMNIC7, HF1-iMNIC11 and HF1-iMNIC12 (*Figure 1—figure supplement 2A* and *Supplementary file 3*). In order to determine whether the fibroblasts acquired motor neuronal identity after *POU5F1(OCT4)*-mediated neural induction, we analyzed the relative mRNA level of MN specification genes *ISL1*, *HB9*, *NKX6.1* and *LHX3* (*Davis-Dusenbery et al., 2014*) in *POU5F1(OCT4)*-infected cells at day 7, day 14 and three iMNIC clones (*Figure 1—figure supplement 2B*). Notably, quantitative reverse transcription polymerase chain reaction (qRT-PCR) analysis revealed that only *ISL1* was dramatically upregulated after *POU5F1(OCT4)* induction (*Figure 1—figure supplement 2B*). Consistently, immunocytochemistry also showed that iMNICs robustly express ISL1, whereas other MN markers (HB9, NKX6.1 and LHX3) or neuronal markers (TUJ1 and MAP2) were not detected (*Figure 1—figure supplement 2C*). For further MN induction and characterization of general features of MNs, we selected HF1-iMNIC2 which expressed *ISL1* gene most highly (*Figure 1—figure supplement 2B*).

To induce iMNICs into mature iMNs, we plated iMNICs on PDL/Laminin-coated plate in MN induction medium containing spinal ventralizing morphogen, sonic hedgehog (SHH) (*Ericson et al., 1996*; *Jessell, 2000*). Most of the cells showed neuronal morphology expressing TUJ1 (>95%), and nearly all TUJI+ cells co-expressed ISL1 at day 28 (*Figure 1—figure supplement 3A*). These cells also expressed mature neuronal marker MAP2 and presynaptic marker synapsin1 (SYN1) (*Figure 1—figure supplement 3B*). However, terminal MN marker HB9 expressing cells were very rare in population (*Figure 1—figure supplement 3C*). These data indicate that *POU5F1(OCT4)* induction potentially contributes to MN induction, but not sufficient for complete reprogramming toward mature MNs.

Given that iMNICs express endogenous *ISL1* gene, we hypothesized that additional transcription factor involved in MN specification might facilitate the conversion of iMNICs toward *bona fide* MNs. We selected six candidate transcription factors including PAX6, *NKX6.1*, *OLIG2*, *LIM1* and *LHX3* (*Davis-Dusenbery et al., 2014*; *Jessell, 2000*) and transduced iMNICs with each gene individually (*Figure 1I*). Also, we employed sphere culture to optimize the culture condition for MN induction. When we plated iMNICs on non-coated dishes for sphere culture, the cells formed clusters and became free-floating iMN spheres in MN induction medium (*Figure 1F*). The iMN spheres were subsequently transferred onto PDL/Laminin-coated dishes for neuronal maturation, then MN-like cells outgrew from the spheres (*Figure 1G*). We could observe robust axon projections and dendritic arborizations, suggesting the neuronal maturation (*Figure 1G'*). Remarkably, the converted neurons exclusively expressed TUJ1 and HB9 only after *LHX3* induction, but no HB9+ cells were observed in other transcription factor infected cells (*Figure 1H*). More than 96% of iMNs co-expressed TUJ1 and

HB9 in sphere culture, while the efficiency was considerably lower in adherent culture (*Figure 1H–J* and *Figure 1—figure supplement 4A*). We applied this to other iMNIC clones, and they were converted into HB9+TUJ1+ iMNs with conversion efficiency of 70 ~ 90% (*Figure 1—figure supplement 4B* and *Supplementary file 4*). Addition of other transcription factors rather impeded neuronal morphology, especially *PAX6* and *OLIG2* known to contribute to the early stage of MN specification even decreased the efficiency of TUJ1+ cells (*Figure 1I* and *Figure 1—figure supplement 4A*). These results demonstrate that *POU5F1(OCT4)* and *LHX3* with our defined culture condition are sufficient to convert cell fate of human fibroblasts into mature iMNs with high purity.

## Direct conversion through self-renewing iMN-intermediate cells (iMNICs) enables large production of iMNs

As described above, iMNICs exhibited neural progenitor-like morphology and proliferative capacity. It has been reported that *POU5F1(OCT4)* can directly convert somatic cells into tripotent neural progenitors (*Mitchell et al., 2014b*). Considering previous report, we evaluated cellular identity of iMNICs whether the cells are similar to neural progenitors. Immunocytochemistry analysis showed that none of iMNICs expressed early neuroectoderm markers (SOX1 and PAX6), neural progenitor markers (SOX2 and NCAM) or MN progenitor marker (OLIG2), and these markers were never detected during the entire neural induction process (*Figure 1—figure supplement 5A*). To determine whether this cell conversion occurred through pluripotent state or neural progenitor state, we evaluated the activation of endogenous expression of pluripotent genes (*POU5F1(OCT4)* and *NANOG)* and neural progenitor marker (*SOX2*) after *POU5F1(OCT4)* induction (*Figure 1—figure supplement 5B*). We found that none of these markers were detected in *POU5F1(OCT)*-induced cells, iMNICs and iMNs as confirmed by qRT-PCR. This result implies that fibroblasts transit neither pluripotent state nor neural progenitor state. Although iMNICs were distinct population from neural progenitors, these cells were highly proliferative cells and capable of being maintained for more than 13 passages. We examined the self-renewal capacity of iMNICs. We evaluated iMNICs at early passage (P2) and late passage (P13). Cellular morphology was very similar at both early and late passages (*Figure 1—figure supplement 5C*). The cells sustained the proliferation rate presenting the mean doubling times (mDT) of 28.4 hr at P2 and 29.4 hr at P13, respectively (*Figure 1—figure supplement 5D*). In addition, we confirmed that >97% of iMNICs expressed proliferative cell marker KI67 (*Figure 1—figure supplement 5E and F*). To verify the continuous capacity of iMNICs to convert into iMNs over multiple passages, we conducted iMN induction using iMNICs at P2 and P13. iMNICs were successfully converted into mature iMNs at both early and late passage (*Figure 1—figure supplement 5G*). Furthermore, to determine whether generation of iMNICs by initial induction of *POU5F1(OCT4)* prior to *LHX3* induction is necessary, we compared the cellular identity of iMNICs emerged after induction of *POU5F1(OCT4)* alone and co-induction of *POU5F1(OCT4)* and *LHX3*. When we infected fibroblasts with *POU5F1(OCT4)* and *LHX3* simultaneously, the majority of infected cells died and iMNIC colonies were not stably maintained for multiple passages after picking (up to 5 passages) (*Figure 1—figure supplement 6A*). Proliferation rate and *ISL1* gene expression were also decreased in *POU5F1(OCT4)/LHX3* induced iMNICs compared to *POU5F1(OCT4)*-induced iMNICs (*Figure 1—figure supplement 6B and C*). In addition, induction of *POU5F1(OCT4)* alone showed higher efficiency in the number of iMNIC colonies (*Figure 1—figure supplement 6D*). These data indicate that the sequential induction of two transcription factors is essential for generating self-renewing iMNICs more efficiently. Together, these results demonstrate that iMNICs generated by *POU5F1(OCT4)* are expandable through self-renewing intermediate state which is distinct from neural progenitor state. Therefore, our method can facilitate the generation of iMNs on large-scale.

## Characterization of iMNs

To monitor the cell fate conversion process in live culture and further characterization of iMNs, we employed reporter system using a lentivirus expressing the green fluorescent protein gene (GFP) under the control of the HB9 promoter (HB9:GFP) (*Marchetto et al., 2008*; *Toli et al., 2015*). After we transfected iMNICs with *LHX3* together with HB9:GFP reporter lentivirus, GFP-positive cells gradually emerged from the cell clusters under SHH stimulation (*Figure 2A*). HB9:GFP+ iMNs were capable of extending long projections (*Figure 2A*). To identify whether these GFP+ iMNs possess the typical characteristics of MNs, we evaluated MN marker expression. Immunocytochemistry

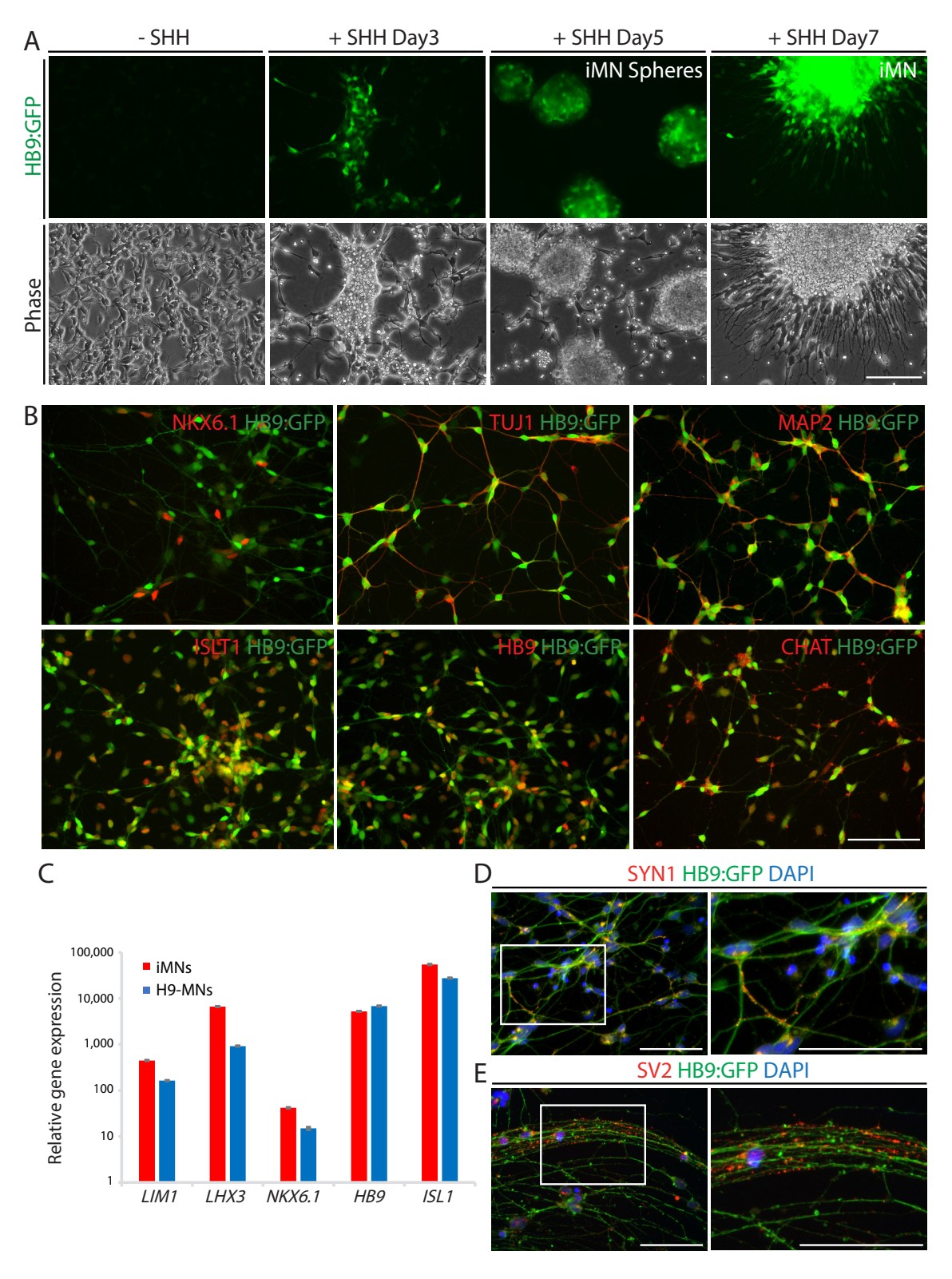

**Figure 2.** iMNs exhibits typical MN characteristics. (**A**) The emergence of HB9:GFP+ cells depending on SHH stimulation during the MN induction. The morphologies of the cells at day 3, 5 and 7 are shown in phase contrast. Scale bars, 125 μm. (**B**) Immunofluorescence images of HB9:GFP+ iMNs co-expressing MN specific markers (NKX6.1, TUJ1, MAP2, ISLT1, HB9 and CHAT). Scale bars, 125 μm. (**C**) MN genes (*LIM1, LHX3, NKX6.1, HB9,* and *ISL1*) are upregulated in iMNs relative to fibroblasts as analyzed by qRT-PCR. MNs derived from H9 ESC (H9-MNs) are used as control. Graphs are presented

*Figure 2 continued on next page*

*Figure 2 continued*

with fold change after normalization by *GAPDH*. Data are presented as mean ± SD, and represent experimental replicates (n = 3). (D-E) Immunofluorescence images of HB9:GFP+ iMNs stained with presynaptic markers, SYN1 (D) and SV2 (E). Zoomed images of the square in the figures show punctate patterns of synaptic terminals. Scale bars, 125 μm. Related data can be found in *Figure 2—figure supplement 1*.

The online version of this article includes the following source data and figure supplement(s) for figure 2:

**Source data 1.** qRT-PCR analysis of MN gene expressions in iMNs.
**Figure supplement 1.** Characterization of iMNs.

analysis showed that HB9:GFP+ iMNs co-expressed MN markers including NKX6.1, ISL1, HB9 and CHAT as well as neuronal markers, TUJ1 and MAP2 (*Figure 2B*). In contrast, tyrosine hydroxylase (TH)-positive dopaminergic neurons were rare in population (<0.6%) (*Figure 2—figure supplement 1A*), and GFAP-positive astrocytes were not detected (*Figure 2—figure supplement 1B*), suggesting that our reprogramming method is specific to MN lineage rather than other neuronal subtypes or neural progenitors. Consistent with the immunocytochemistry results, the endogenous mRNA levels of MN marker genes including *LIM1, LHX3, NKX6.1, HB9* and *ISL1* were upregulated in iMNs relative to fibroblasts (*Figure 2C*). Furthermore, we could observe the synaptic activities of HB9:GFP+ neurites expressing presynaptic markers, SYN1 and synaptic vesicle protein 2 (SV2) (*Figure 2D,E* and *Figure 1—figure supplement 5H*). We next confirmed the transgene silencing in iMNs. Exogenous expression of transgenes *POU5F1(OCT4)* and *LHX3* are dramatically decreased in iMNs, as examined by qRT-PCR (*Figure 2—figure supplement 1C and D*). Together, these results demonstrate that iMNs exhibit the typical characteristics of MNs.

## iMNs exhibit electrophysiological properties

To determine whether iMNs are fully mature, we identified electrophysiological properties of iMNs by conducting whole-cell patch clamp recording. 29 out of 48 tested cells (60.42%) exhibited at least one action potential (AP) firing, while 19 cells (39.58%) were unresponsive to depolarizing current injection (*Figure 3—figure supplement 1A*). Among the cells showing AP (29 cells), about half of the cells (14 cells) generated a multiple number of AP firings and 8 out of 14 cells also showed spontaneous firing at resting membrane potential (*Figure 3—figure supplement 1A*). Those cells displaying multiple APs were further grouped based on the presence of spontaneous firing (group 1 vs. group 2). We first tested passive properties of neuronal membrane by injecting hyperpolarizing current. Both resting membrane potential and input resistance were comparable between group 1 and group 2 (*Figure 3A–D*). Next, we compared the active properties of iMNs between the two groups by injecting depolarizing current (*Figure 3E*). Likewise, there were no significant differences in AP threshold, frequency, amplitude, half-width and neuronal excitability between the two groups (*Figure 3F–J*), suggesting that these cells are functionally matured regardless of spontaneous firing at resting membrane potential. In voltage clamp mode, depolarizing voltage steps also induced fast inward currents. These currents were completely blocked by the bath-application of tetrodotoxin (TTX). We confirmed that these inward currents were elicited by TTX-sensitive voltage-gated sodium channels (*Figure 3K–M*). Together, these data indicate that iMNs are electrophysiologically mature.

## iMNs form neuromuscular junctions with myotubes

To evaluate the in vitro functionality of iMNs, we investigated whether iMNs possess the ability to form neuromuscular junctions (NMJs) with muscles, which is the key feature of spinal MNs. We differentiated mouse myoblasts C2C12 into multinucleated myotubes and co-cultured with iMNs. Extensive axons of HB9:GFP+ iMNs projected along the myotubes (*Figure 4A*). Noticeably, SV2-positive vesicles were seen in HB9:GFP+ iMNs and enriched at NMJs showing puncta-like structures (*Figure 4A*). At the sites of contacts with myotubes, clusters of postsynaptic acetylcholine receptors (AChR) on the surface of myotubes were detected by immunocytochemistry with AChR antibody (*Figure 4B*) and α-bungarotoxin conjugated with Alexa 555 (α-BTX) (*Figure 4C*). The formation of NMJs was also observed with iMNs generated from iMNICs at late passage (*Figure 1—figure supplement 5I*). Importantly, we could observe the rhythmic contraction of myotubes when co-cultured with iMNs. The contractions were blocked by antagonist of nicotinic AChR, curare, indicating that the myotubes contract depending on the activities of iMNs (*Video 1*). These data demonstrate that

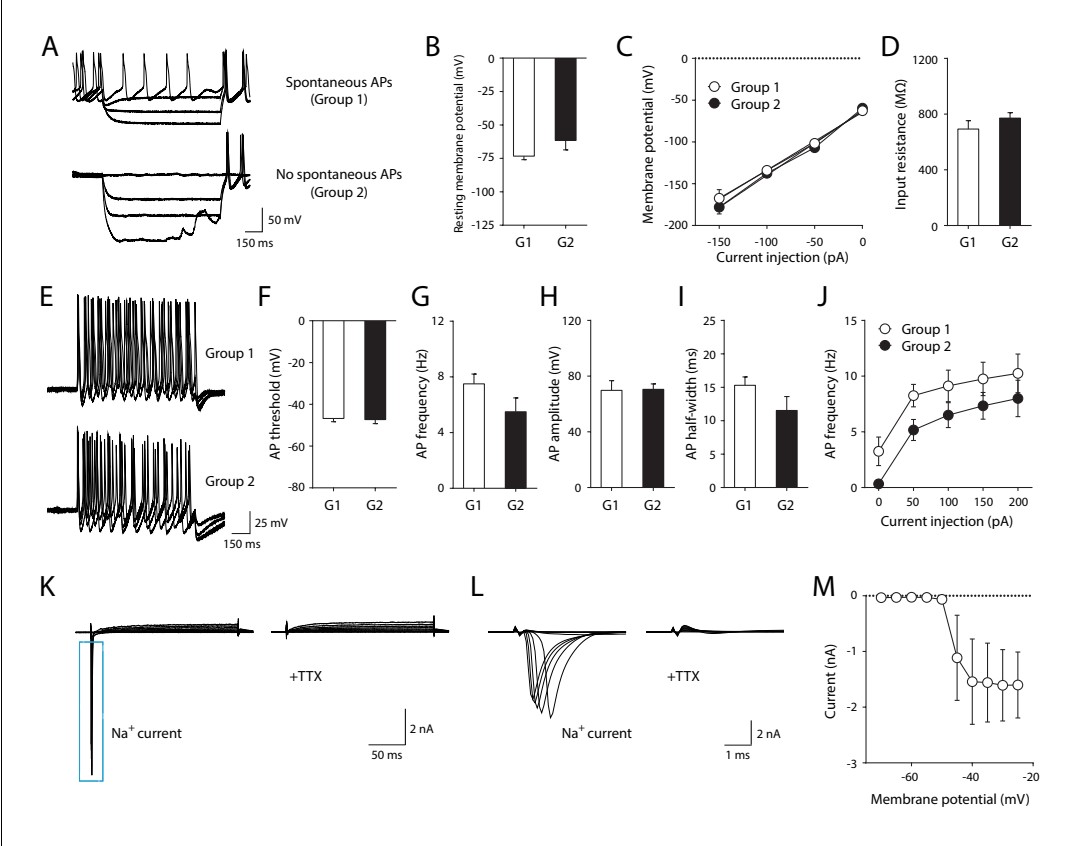

**Figure 3.** Electrophysiological properties of iMNs. (**A**) Representative recording traces of iMNs by hyperpolarizing current injection under current clamp-mode. (**B**) Resting membrane potential (Group1, −73.29 ± 2.65 mV, n = 8; Group2, −61.57 ± 7.04 mV, n = 6; unpaired t-test, $t_{12}$ = 1.7330, p=0.1086). (**C**) Current (I) - voltage (V) relationship in iMNs by hyperpolarizing current step (repeated measures two-way ANOVA, n = 8 for Group1, n = 6 for Group 2; group effect, $F_{1,\,12}$ = 0.5074, p=0.4899; interaction, $F_{3,\,36}$ = 0.6432, p=0.5923). (**D**) Input resistance of iMNs (Group1, 692.89 ± 60.60 MΩ, n = 8; Group2 = 771.43 ± 39.09 MΩ, n = 6; unpaired t-test, $t_{12}$ = 1.0050, p=0.3349). (**E**) Representative recording traces of iMNs by depolarizing current injection under current clamp-mode. (**F**) AP threshold (Group1, −46.76 ± 1.58 mV, n = 8; Group2, −47.30 ± 1.94 mV, n = 6; unpaired t-test, $t_{12}$ = 0.2187, p=0.8305). (**G**) AP frequency (Group1, 7.50 ± 0.71 Hz, n = 8; Group2, 5.50 ± 0.99 Hz, n = 6; unpaired t-test, $t_{12}$ = 1.6920, p=0.1165). (**H**) AP amplitude (Group1, 69.81 ± 6.97 mV, n = 8; Group2, 70.55 ± 3.91 mV, n = 6; unpaired t-test, $t_{12}$ = 0.0840, p=0.9345). (**I**) AP half-width (Group1, 15.30 ± 1.25 ms, n = 8; Group2 = 11.53 ± 2.09 ms, n = 6; unpaired t-test, $t_{12}$ = 1.6390, p=0.1271). (**J**) Neuronal excitability of iMNs (repeated measures two-way ANOVA, n = 8 for Group1, n = 6 for Group 2; group effect, $F_{1,\,12}$ = 2.9410, p=0.1120; interaction, $F_{4,\,48}$ = 0.0774, p=0.9888). (**K–L**) Representative TTX-sensitive sodium currents (**K**) of iMNs (TTX, 1 µM). (**L**) Zoomed-in view of TTX-sensitive sodium currents shown in (**K**). (**M**) Averaged current (I)/voltage (V) curve of sodium currents (n = 11). Mean ± SEM is used for all the data described in the figure. Related data can be found in *Figure 3—figure supplement 1*.

The online version of this article includes the following figure supplement(s) for figure 3:

**Figure supplement 1.** Physiological properties of iMNs.

mature iMNs exhibit mature functionality such as the formation of NMJs. To evaluate our iMN direct conversion method, we generated iMNICs and iMNs from additional human fibroblast line, HF2 (*Figure 4—figure supplement 1A*). HF2-derived iMNs (HF2-iMNs) showed typical characteristics of MNs and functions, such as synaptic activities and NMJ formation (*Figure 4—figure supplement 1B–D*). We could establish five iMNIC clones from HF2; HF2-iMNIC1, HF2-iMNIC2, HF2-iMNIC4, HF2-iMNIC6 and HF2-iMNIC8) (*Figure 4—figure supplement 1E*). These iMNIC clones highly expressed *ISL1* gene, and the cells were converted into HB9+iMNs after MN induction (*Figure 4—figure supplement 1F* and *Figure 4—figure supplement 1G*). Taken together, these results confirm that iMN conversion method is applicable to other lines of human adult fibroblasts.

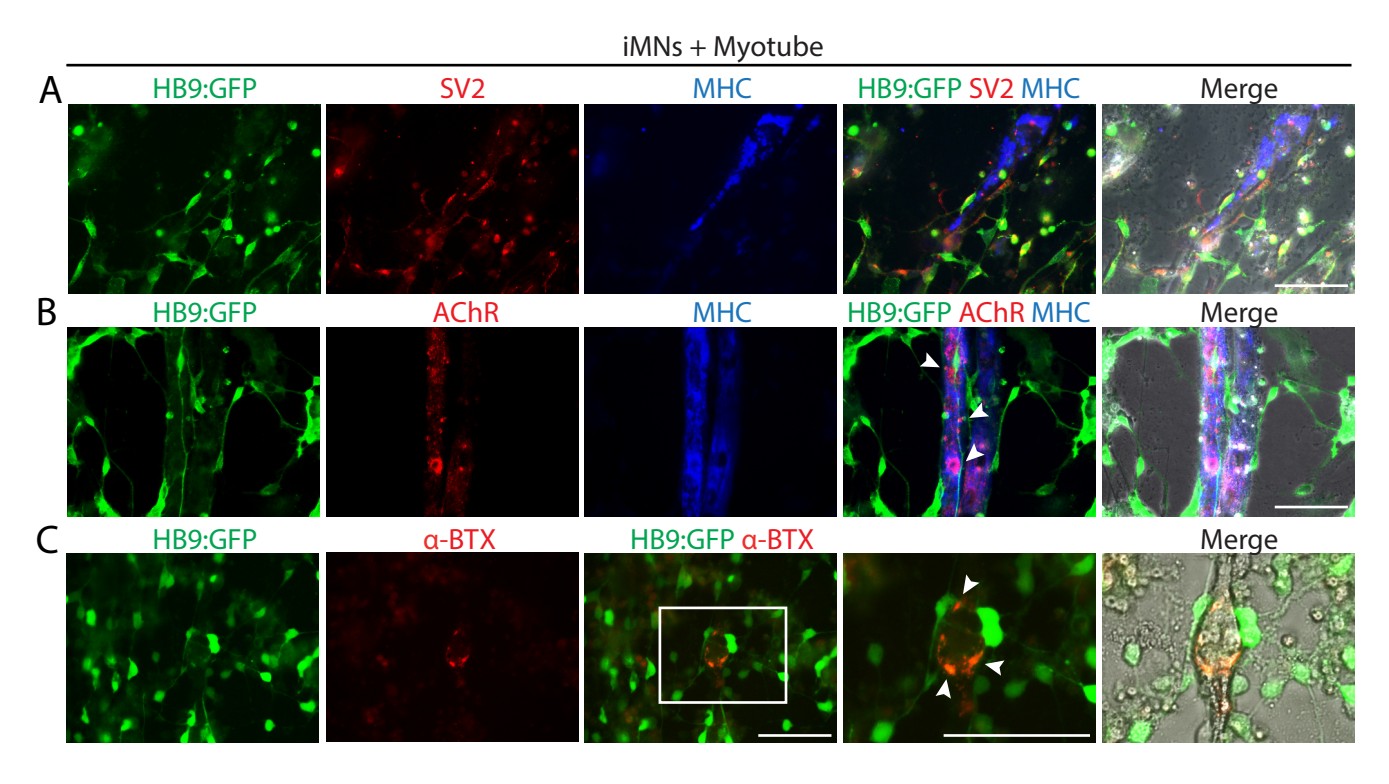

**Figure 4.** iMNs form neuromuscular junctions with myotubes. (A–C) Representative immunofluorescent analysis of neuromuscular junctions (NMJs) formed between HB9:GFP+ iMNs and C2C12 myotubes in co-culture. Presynaptic axons and post-synaptic acetylcholine receptor (AChR) are stained with antibodies against SV2 (A) and AChR (B), and the multinucleated myotubes are stained with myosin heavy chain (MHC). (C) NMJs are labeled with α-bungarotoxin conjugated with Alexa 555. Zoomed images of the square in (C) show the NMJs, indicated by white arrows. Scale bars, 75 μm. Related data can be found in *Figure 4—figure supplement 1*.

The online version of this article includes the following figure supplement(s) for figure 4:

**Figure supplement 1.** Generation of iMNs from additional human fibroblasts (HF2).

## Global transcriptional profiles of iMNICs and iMNs by RNA-sequencing

To examine the identity of iMNICs and iMNs, we performed RNA-sequencing (RNA-seq) analysis to compare global gene expression profiles of iMNICs and iMNs to that of parental fibroblasts (HF1 and HF2), fetal NPCs (NPC) (*Kim et al., 2009a*), wild type motor neurons (wtMNs) derived from human ESC (wtMN-1) (*Amoroso et al., 2013*) and iPSC (wtMN-2) (*Ng et al., 2015*) and fetal spinal cord (Fetal-SC) (*Kumamaru et al., 2018*). Heatmap analysis demonstrated that the global gene expression patterns of iMNICs and iMNs were similar to wtMNs, but distinct from fibroblasts and fetal NPCs (*Figure 5A*). The hierarchical clustering and principal component analysis (PCA) showed that iMNICs, iMNs and wtMNs are tightly corre-lated, whereas distinct from NSCs (*Figure 5B and C*). The 1st principal component (PC1)

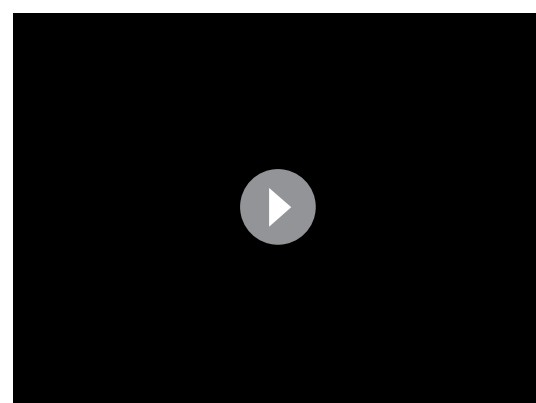

**Video 1.** Contraction of C2C12 myotubes co-cultured with iMNs. C2C12 myotubes started rhythmic contraction after co-culture with iMN for 3 weeks. To block the acetylcholine receptor on the myotubes specifically, 100 μM curare (final concentration) was added to the culture.
https://elifesciences.org/articles/52069#video1

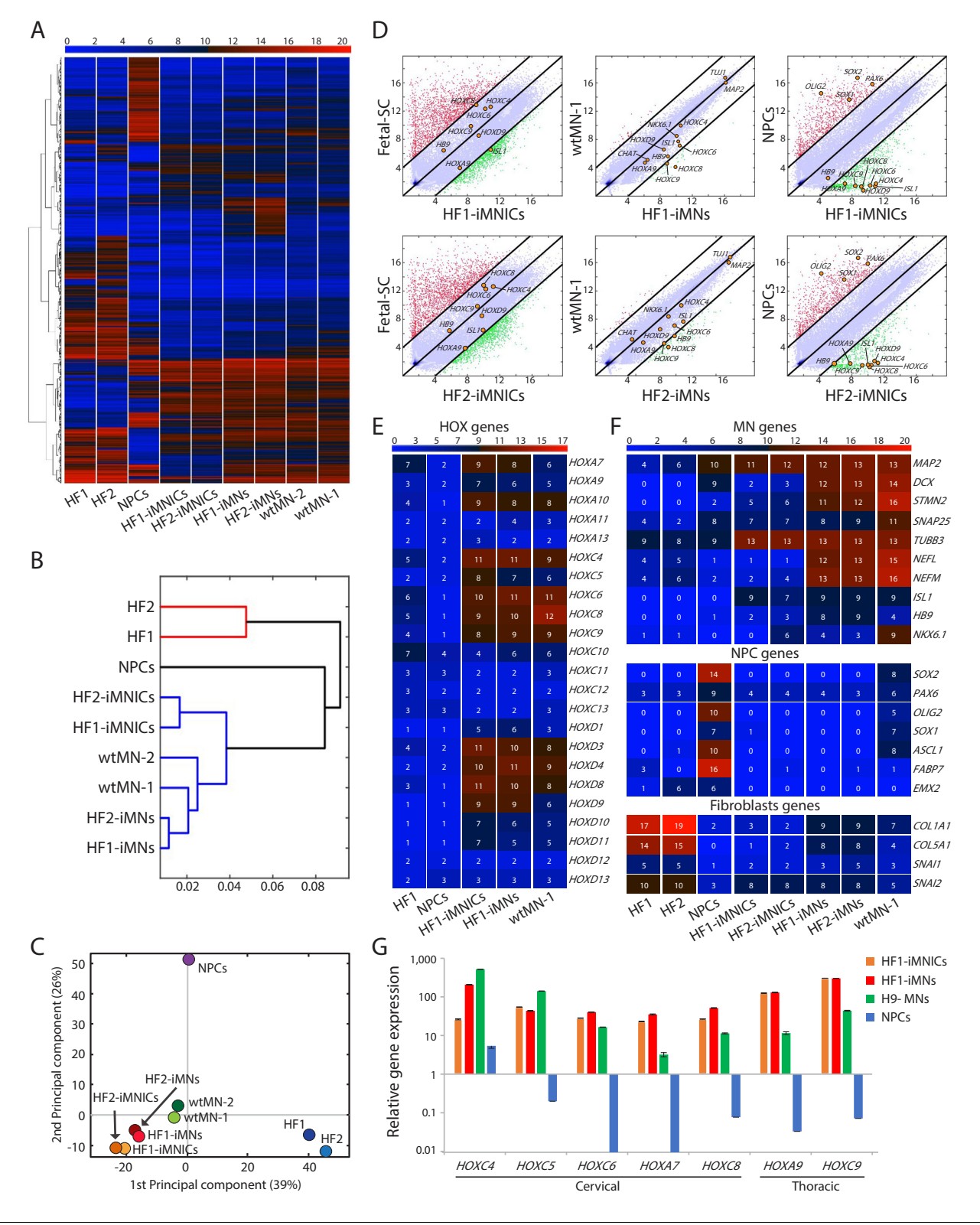

**Figure 5.** Global gene expression profiles of iMNICs and iMNs. (**A**) Heatmap analysis of global gene expression profiles in HF-1, HF-2, NPCs, HF1-iMNICs, HF2-iMNICs, HF1-iMNs, HF2-iMNs, wtMN-1 and wtMN-2 as determined by RNA-seq. The color bar codifies the gene expression in log₂ scale. Red indicates upregulated genes and blue indicates downregulated genes. (**B**) Hierarchical clustering of HF2, HF1, NPCs, HF2-iMNICs, HF1-iMNICs, wtMN-2, wrMN-1, HF2-iMNs and HF1-iMNs. (**C**) PCA of global gene expression in HF1, HF2, HF1-iMNICs, HF2-iMNICs, HF1-iMNs, HF2-iMNs, NPCs,

*Figure 5 continued on next page*

Figure 5 continued

wtMN-1 and wtMN-2. (D) Pairwise scatter plots of global gene expression of Fetal-SC vs HF1-iMNICs/HF2-iMNICs, wtMN-1 vs HF1-iMNs/HF2-iMNs and NPCs vs HF1-iMNICs/HF2-iMNICs. The HOX genes (*HOXC4, HOXC6, HOXC8, HOXC9, HOXA9* and *HOXD9*), MN genes (*HB9, ISL1, NKX6.1 MAP2, TUJ1* and *CHAT*), fibroblast genes (COL5A1 and COL1A1) and NPC genes (*SOX2, OLIG2, PAX6* and *SOX1*) are highlighted with yellow circles. The black lines indicate the boundaries of 4-fold changes in log$_2$ scale. (E and F) Heatmap analysis of (E) HOX genes, (F) MN genes, NPC genes and fibroblast genes in HF1, HF2, NPCs, HF1-iMNICs, HF2-iMNICs, HF1-iMNs, HF2-iMNs and wtMN-1. The color bar codifies the gene expression in log$_2$ scale. Red indicates upregulated genes and blue indicates downregulated genes. (G) qRT-PCR analysis of mRNA expression level for cervical *HOX* genes (*HOXC4, HOXC5, HOXC6, HOXA7* and *HOXC8*) and thoracic *HOX* genes (*HOXA9* and *HOXC9*) in HF1-iMNINs, HF1-iMNs, H9-MNs and NPCs. Bars represent fold changes relative to fibroblasts after normalization to *GAPDH*. Data are presented as means ± SD (n = 3). Related data can be found in *Figure 5—figure supplement 1*.

The online version of this article includes the following source data and figure supplement(s) for figure 5:

**Source data 1.** qRT-PCR analysis of cervical and thoracic *HOX* genes.
**Figure supplement 1.** RNA-seq analysis of iMNICs and iMNs.

captures 39% of the gene expression variability and the 2nd principal component (PC2) captures 26% of the variability. Moreover, pairwise scatter plots showed high similarity between iMNICs vs fetal spinal cord and iMNs vs wtMNs, especially in *HOX* genes (*HOXC4, HOXC6, HOXC8, HOXC9, HOXA9* and *HOXD9*) and MN-enriched genes (*HB9, ISL1, NKX6.1, TUJ1, MAP2* and *CHAT*) (*Figure 5D*). In contrast, we could observe low similarity between iMNICs vs NPC and parental fibroblasts vs iMNICs/iMNs, especially NPC marker genes (*SOX2, OLIG2, PAX6* and *SOX1*) and fibroblast genes (*COL5A1* and *COL1A1*) (*Figure 5D* and *Figure 5—figure supplement 1A*). Furthermore, we could observe high level of similarity in *HOX* gene clusters and MN genes in our iMNICs and iMNs compared to wtMNs, which were not expressed in NPCs (*Figure 5E,F* and *Figure 5—figure supplement 1B*). In addition, iMNICs and iMNs expressed similar level of HOXC clusters and *ISL1* with wtMN-1, but *SOX2* expression was not detected in iMNICs and iMNs (*Figure 5—figure supplement 1D*). In contrast, the expression of NPC genes (*SOX2, PAX6, OLIG2, ASCL1, FABP7* and *EMX*), fibroblast-specific genes (*COL1A1, COL5A1, SNA1* and *SNAI2*) and pluripotent genes (*NANOG, POU5F1 (OCT4)* and *TDGF1*) were not detectable in iMNICs and iMNs (*Figure 5F* and *Figure 5—figure supplement 1C*). To validate the RNA-sequencing data, we conducted qPCR to analyze mRNA expression of *HOX* genes in our iMNICs and iMNs. Consistent with RNA-sequencing data, cervical and thoracic spinal cord specific genes (*HOX*4-9) were upregulated in iMNICs and iMNs as confirmed by qRT-PCR (*Figure 5G*). Together, these results indicate that iMNs generated from two fibroblast lines acquired motor neuronal identity and showed a high degree of similarity with wtMNs or fetal spinal cord tissues.

## Therapeutic potential of iMNs in spinal cord injury (SCI) model in vivo

In order to examine the in vivo functionality and therapeutic effects of iMNs, we transplanted HB9: GFP+ iMNs into adult rat SCI models (*Figure 6A*). We induced compressive damage to thoracic vertebrae 9 (T9) of the spinal cord and injected 1 × 10$^6$ iMNs into the upper (T8) and lower (T10) vertebrate after 1 week of injury. We could observe the engraftment of transplanted HB9:GFP+ iMNs in the spinal cord (*Figure 6B*). To evaluate the cellular features of iMNs in vivo, the spinal cord tissue was immunostained with neuronal marker (TUJ1) and oligodendrocyte marker (MBP). Most of the GFP+ cells co-expressed TUJ1 (*Figure 6C–E*) and were surrounded by host myelinating MBP+ oligodendrocytes (*Figure 6F–H*). To confirm the tissue recovery of the injury site, we conducted hematoxylin and eosin (H and E) staining of sagittal sections of the spinal cord after 8 weeks of transplantation to detect the cavity size. Transplanted tissue show less cavity of injured site compared to control (*Figure 6I and J*). Next, we evaluated the motor function recovery of the hind limbs by measuring Basso–Beattie–Bresnahan (BBB) scores for 8 weeks in rat SCI models that were treated either with PBS (control) (*n* = 6) or with iMNs transplantation (*n* = 10). iMN-transplanted rats were improved in BBB scores from week 4 that persisted steadily through week showing significant improvement in locomotor recovery (*Figure 6K,L* and *Video 2*). To assess the risk of tumor formation, iMNICs and iMNs were subcutaneously transplanted into immune-deficient nude mice and no tumors were observed for 12 months of experiment period (n = 4) (*Figure 6—figure supplement 1A*). These results showed that iMN transplantation significantly improved functional recovery after SCI.

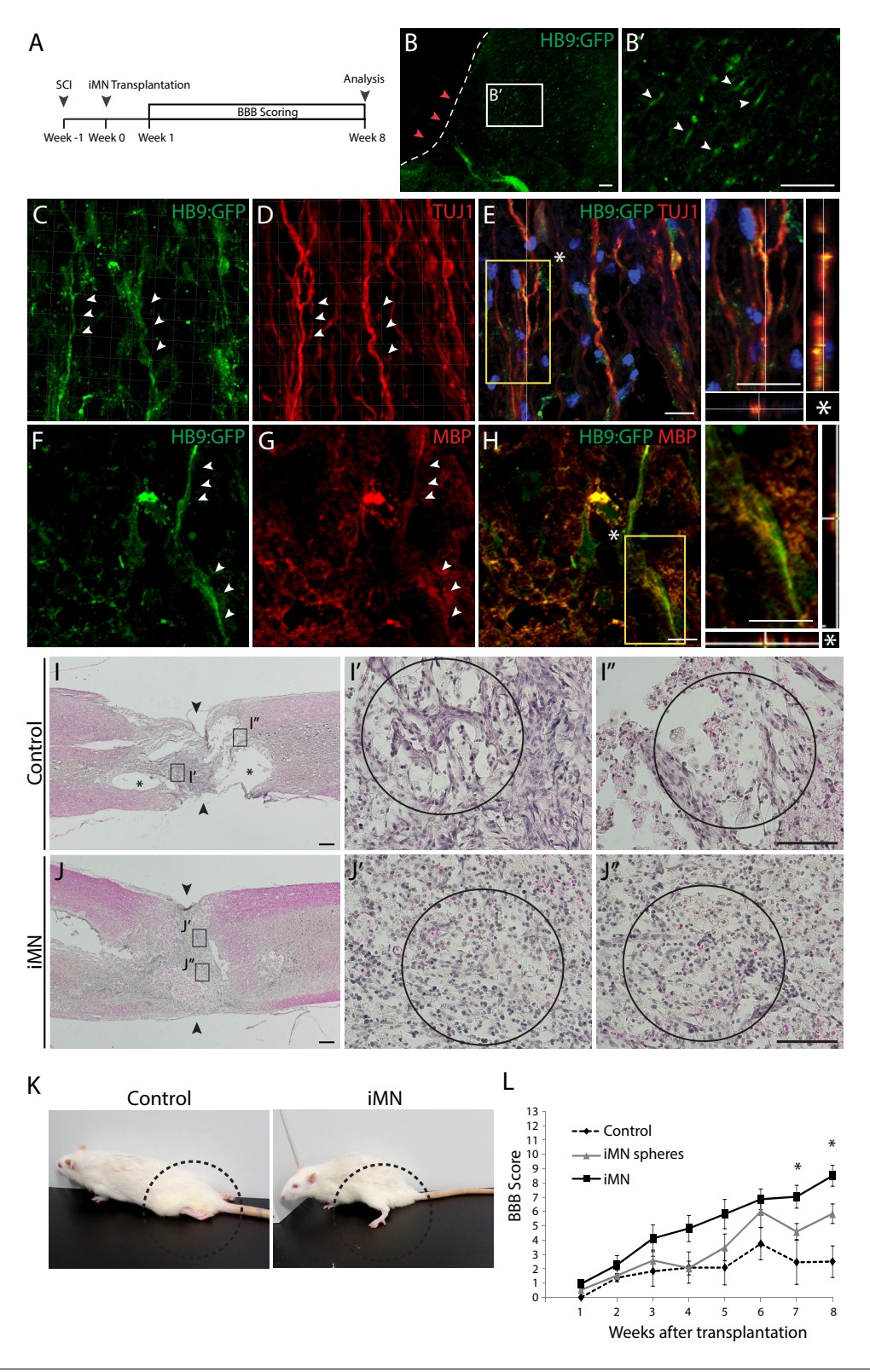

**Figure 6.** Therapeutic effects of iMNs in rat spinal cord injury model in vivo. (**A**) The experimental scheme of in vivo study. (**B**) Immunofluorescence image of transplanted iMNs in sagittal section of spinal cord. (**B'**) Zoomed image presents integration of transplanted cells. Scale bars, 125 µm. (**C–E**) Confocal images of iMNs after transplantation. HB9:GFP+ cells co-express neuronal marker, TUJ1 (white arrowheads). (**F–H**) iMNs are surrounded by host MBP+ myelinating cells forming ensheathment (white arrowheads). Scale bars, 30 µm. (I-J. H and E staining analysis of spinal cords after 8 weeks of

*Figure 6 continued on next page*

*Figure 6 continued*

transplantation (I; Control, J; iMN-transplanted) (I,J; Scale bars, 400 µm, I',I'',J',J''; Scale bars, 125 µm). (K) The position of hindlimbs in control rat and iMN-transplanted rat after 8 weeks of transplantation. (L) BBB score evaluation of hindlimbs during 8 weeks after transplantation. Data are presented as mean ± SD, and represent experimental replicates (Control; n = 6, iMN sphere; n = 9, iMN; n = 10). *p<0.05, one-way ANOVA. Related data can be found in *Figure 6—figure supplement 1*.

The online version of this article includes the following figure supplement(s) for figure 6:

**Figure supplement 1.** Tumor formation analysis of iMNICs and iMNs in vivo.

## Discussion

In this study, we established an advanced direct conversion strategy to generate induced motor neurons (iMNs) from human fibroblasts in large-scale with high purity, thereby providing a cell source for the treatment of spinal cord injury (SCI). Previously, we succeeded in generating oligodendrocyte progenitor cells (OPCs) from somatic cells by *POU5F1(OCT4)* (*Kim et al., 2015*). In line with this, we hypothesized that combination of *POU5F1(OCT4)* and defined supplements may convert cell fate toward motor neurons (MNs) which is developmentally derived from the same origin with OPCs, pMN progenitors of the ventral spinal cord (*Ravanelli and Appel, 2015*). *POU5F1(OCT4)* could efficiently generate TUJ1-positive neuronal cells under MN induction conditions, however, most of the cells could not reach to HB9 positive mature MNs. Nonetheless, we found that endogenous expression of *ISL1*, an important transcription factor for MN specification (*Liang et al., 2011*), was activated in fibroblasts after *POU5F1(OCT4)* induction under our defined culture condition. Based on this observation, we assumed that an additional transcription factor is required for complete iMN conversion. By screening the MN specification factors, we found that inclusion of *LHX3* significantly increased the reprogramming efficiency of HB9 positive iMNs. This result is consistent with the previous report that the *LIM* complex composed of ISL1 and LHX3 specify spinal MNs by inducing MN gene *HB9* in development (*Lee et al., 2012*). To our knowledge, this is the first time showing that *POU5F1(OCT4)* can initiate cellular reprogramming toward MN lineage by activating *ISL1*. This is correlated with previous reports that *LIM* homeodomain transcription factor *ISL1* has been detected as POU5F1(OCT4) targets in human pluripotent cells (*Boyer et al., 2005*; *Jung et al., 2010*).

This result has an important implication on how we minimized the transcription factors for generating iMNs. Previously, iMNs were generated by either eight transcription factors (*ASCL1, BRN2, MYT1L, LHX3, HB9, ISL1, NGN2,* and *NEUROD1*) (*Son et al., 2011*) or four transcription factors (*NGN2, SOX11, ISL1,* and *LHX3*) (*Liu et al., 2016*). In ; contrast to previous methods, we identified the minimal two transcription factors, *POU5F1(OCT4)* and *LHX3*, for generating iMNs, so that our iMNs would be safer than multiple transcription factors-derived iMNs for therapeutic applications in terms of lower chance of viral integrations (*Kim et al., 2009b*). Moreover, previous studies have not verified the in vivo functionality or therapeutic potential of human iMNs.

In addition, RNA-sequencing analysis revealed that our iMNs exhibited transcriptional profiles of motor neuronal identity similar with wtMNs. Importantly, iMNICs and iMNs expressed spinal cord specific *HOX5-9* gene clusters, whereas NPC genes were not detected in these cells. The ability to generate autologous iMNs with motor neuronal identity can facilitate disease modeling and cell-based therapy for MN diseases or spinal cord disorders. A critical prerequisite for achieving these goals is obtaining the pure population of subtype-specific neurons in high number. However, previous methods have not been feasible since directly converted neurons are terminally differentiated cells that are not scalable. This limitation could be overcome by generating expandable neural progenitors; however, neural progenitors could produce a mixture of heterogeneous neurons (*Lujan et al., 2012*; *Ring et al., 2012*; *Thier et al., 2012*). Herein, sequential introduction of two transcription factors enabled the large production of pure iMNs. We could obtain self-renewing cell line iMNICs after *POU5F1(OCT4)* induction by clonal reprogramming method as we have demonstrated previously in generating OPCs from fibroblasts (*Kim et al., 2015*). iMNICs were proliferative, expandable and retained conversion capacity to mature iMNs for long-term culture (>13 passages); however, they were distinct from neural progenitors. Therefore, our method is 'direct conversion' rather than indirect conversion or redifferentiation through pluripotent state or neural progenitor state.

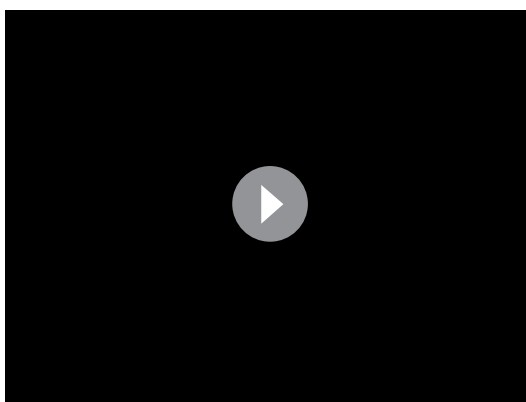

**Video 2.** Functional recovery after iMN transplantation in rat SCI model. Transplantation of iMNs facilitated the recovery of hindlimb motor functions after 8 weeks in rat SCI model. PBS injected control group exhibited no hindlimb movement.
https://elifesciences.org/articles/52069#video2

Previously, Mitchell et al. generated iNPC from *POU5F1(OCT4)*-induced fibroblasts by culturing the cells in reprogramming media (RM) for 8 days before inducing in classical NPC media (*Mitchell et al., 2014a*; *Mitchell et al., 2014b*). They reported that RM step is essential to induce SOX2+ NPCs after *POU5F1(OCT4)* induction. In contrast, SOX2 was not activated in *POU5F1 (OCT4)*-infected cells and failed to generate iNPCs when cultured directly in NPC media (*Mitchell et al., 2014a*). Consistent with previous report by Mitchell et al., our iMNICs which were generated without RM step did not express neural progenitor marker (*Mitchell et al., 2014a*; *Mitchell et al., 2014b*). This finding indicates that extracellular environment and timing for appropriate stimuli are critical for cell identification of *POU5F1(OCT4)*-induced cells.

Several studies have reported that transplantation of MNs or MN precursors differentiated from pluripotent stem cells (PSCs) showed functional benefits to injured spinal cord (*Rossi et al., 2010*; *Wyatt et al., 2011*). However, translation of these cells to the clinic is limited due to the challenges including differentiation efficiency, scalability, purity, and especially tumorigenecity from undifferentiated cells (*Nori et al., 2015*). Moreover, transplantation of MN precursors remain poorly defined because of stem cell heterogeneity (*Trawczynski et al., 2019*). In this study, we could obtain pure iMNs in large number and found that hindlimb functional recovery depends on maturity and purity of iMNs. When we transplanted selected HB9+ mature iMNs, BBB score was higher compared to transplanting iMN spheres. We also observed that our iMNs exhibited integration properties when transplanted into injured spinal cord. Transplanted HB9+ iMNs still expressed neuronal markers and interacted with host neural cells such as oligodendrocytes in vivo. We could observe ensheathment by myelinating host oligodendrocytes surrounding our iMNs. Notably, transplantation of iMNs contributed histological and functional recovery in SCI model without tumor formation, and this is the first report demonstrating the therapeutic effects of human iMNs in vivo.

In conclusion, this proof-of-concept study shows that our functional iMNs can be employed to cell-based therapy as an autologous cell source. iMNs possessed spinal cord motor neuronal identity and exhibit hallmarks of spinal MNs such as neuromuscular junction formation capacity and electrophysiological properties in vitro. Importantly, transplantation of iMNs improved locomotor function in rodent SCI model without tumor formation. Although further investigation on mechanism responsible for cell fate conversion may be needed, our strategy is a safer and simpler methodology that may provide new insights to develop personalized stem cell therapy and drug screening for MN diseases or spinal cord disorders.

## Materials and methods

**Key resources table**

| Reagent type (species) or resource | Designation | Source or reference | Identifiers | Additional information |
|---|---|---|---|---|
| Cell line (*Homo sapiens*) | H9 | WiCell | Cat.# WA09 RRID:CVCL_9773 | Control |
| Cell line (*Homo sapiens*) | NPCs | DOI: 10.1038/ nature08436, *Kim et al., 2009a* | | Control |

*Continued on next page*

Continued

| Reagent type (species) or resource | Designation | Source or reference | Identifiers | Additional information |
|---|---|---|---|---|
| Cell line (*Homo sapiens*) | HF1 (Female) | DOI: 10.4172/2157-7633.1000336 *Singhal et al., 2016* | | Parental cells |
| Cell line (*Homo sapiens*) | HF2 (Male) | DOI: 10.4172/2157-7633.1000336. *Singhal et al., 2016* | | Parental cells |
| Antibody | Anti-HB9 (Mouse Monoclonal) | DSHB | Cat.# 81.5C10 RRID:AB_2145209 | ICC(1:100) |
| Antibody | Anti-ISLT1/2 (Mouse Monoclonal) | DSHB | Cat.# 39.4D5 RRID:AB_2314683 | ICC(1:300) |
| Antibody | Anti-SV2 (Mouse Monoclonal) | DSHB | Cat.# SV2 RRID:AB_2315387 | ICC(1:100) |
| Antibody | Anti-beta III Tubulin (Rabbit polyclonal) | Millipore | Cat.# ab18207 RRID:AB_444319 | ICC(1:500) |
| Recombinant DNA reagent | Lenti-HB9:GFP (plasmid) | Addgene | ID# 37080 RRID:Addgene_37080 | |
| Recombinant DNA reagent | PAX2 (plasmid) | Addgene | ID# 12260 RRID:Addgene_12260 | |
| Recombinant DNA reagent | VSVG (plasmid) | Addgene | ID# 8454 RRID:Addgene_8454 | |
| Recombinant DNA reagent | Lenti-hOCT4 | Addgene | ID# 130692 RRID:Addgene_130692 | |
| Recombinant DNA reagent | Lenti-hLHX3 | Addgene | ID# 120456 RRID:Addgene_120456 | |
| Recombinant DNA reagent | Lentiviral backbone | Addgene DOI: 10.1038 *Warlich et al., 2011* | ID# 12252 RRID:Addgene_12252 | |
| Peptide, recombinant protein | SHH | Peprotech | Cat.# 100–45 | 100 ng/ml |
| Peptide, recombinant protein | bFGF | Peprotech | Cat.# 100-18B | 20 ng/ml |
| Peptide, recombinant protein | EGF | Peprotech | Cat.# AF-100–15 | 10 ng/ml |
| Peptide, recombinant protein | IGF-1 | Peprotech | Cat.# 100–11 | 10 ng/ml |
| Peptide, recombinant protein | NT3 | Peprotech | Cat.# 450–03 | 10 ng/ml |
| Peptide, recombinant protein | BDNF | Peprotech | Cat.# 450–02 | 10 ng/ml |
| Peptide, recombinant protein | GDNF | Peprotech | Cat.# 450–10 | 10 ng/ml |
| Peptide, recombinant protein | CNTF | Peprotech | Cat.# 450–13 | 10 ng/ml |

*Continued*

| Reagent type (species) or resource | Designation | Source or reference | Identifiers | Additional information |
|---|---|---|---|---|
| Other | α-Bungarotoxin | Invitrogen | Cat.# B35451 | 1:200 |
| Software, algorithm | Matlab | | RRID:SCR_001622 | |

## Study plan and ethics

This study was designed to establish an advanced direct lineage reprogramming strategy to generate autologous iMNs from human fibroblasts by overexpressing *POU5F1(OCT4)* and *LHX3*. We further investigated the therapeutic effects of iMNs for treating traumatic spinal cord injury using rodent spinal cord injury model. The experiments were repeated at least three times, and the replicates are indicated in each figure and legend. N values represent the number of animals in the experiment. Quantifications are analyzed by randomly imaging the positions of culture dishes to determine the reprogramming efficiency. Animals used for experiments were assigned randomly to groups. The behavioral tests were performed by blinded observer. The experiments were carried out in accordance with documented standards of the Institutional Review Board of Ulsan National Institute of Science and Technology (UNIST) (UNISTIRB-15–17 C) for human cell experiments. All animal experimental and surgical procedures on animals were performed in accordance with institutional protocols approved by the Institutional Animal Care and Use Committee of Yonsei University College of Medicine (2015–0327) for rat experiments, and the Institutional Animal Care and Use Committee of Ulsan National Institute of Science and Technology (UNIST) (UNISTIACUC-17–34) for mouse experiments.

## Cell culture

Human adult fibroblast lines, HF1 and HF2 were obtained from surgical resectates (*Singhal et al., 2016*), which have been obtained with the informed consent (*Haridass et al., 2009*). Parental fibroblasts were maintained in fibroblast medium (high-glucose DMEM (GIBCO) containing 10% fetal bovine serum (FBS) (GIBCO), 1% penicillin/streptomycin, 1 mM l-glutamine, 1% non-essential amino acids (NEAA) (GIBCO), and 0.1 mM β-mercaptoethanol (GIBCO)). Fibroblast information is summarized in *Supplementary file 3*. 293 T cells used for virus production were maintained in high-glucose DMEM containing 10% FBS, penicillin/streptomycin, l-glutamine. Human ESCs H9 (WiCell) were maintained on irradiated CF1 mouse feeder layers in human ESC medium (knockout DMEM (Invitrogen) supplemented with 20% knockout serum replacement (GIBCO), 1 mM l-glutamine, 1% non-essential amino acids, 0.1 mM β-mercaptoethanol, 1% penicillin/streptomycin and 5 ng/ml human basic fibroblast growth factor (bFGF) (Peprotech). Human fetal NPCs (*Kim et al., 2009a*) were cultured in DMEM/F12 (GIBCO) supplemented with N2 (GIBCO), penicillin/streptomycin, 20 ng/ml bFGF (Peprotech) and 8 μg/ml heparin and 10 ng/ml. Cell lines were authenticated using PCR detection kit. The cells were routinely tested for mycoplasma contaminations, and we used mycoplasma-free cells for experiments.

## Virus construction and production

The cDNAs of candidate MN transcription factors were amplified by PCR, and individually subcloned into the lenitiviral vector backbone (Addgene ID#12252) (*Warlich et al., 2011*). Plasmids carrying human *OCT4* and *LHX3* cDNAs were purchased from Addgene (OCT4; ID#130692, LHX3; ID#120456). Also, plasmid carrying the HB9 promoter and GFP protein were purchased from Addgene (ID#37080) (*Marchetto et al., 2008*).

The viruses were produced and harvested as previously described (*Zaehres and Daley, 2006*). In short, 293 T cells were seeded at 40–50% confluency on 10 cm plates prior to transfection. Individual transfer plasmids, packaging plasmid (PAX2, Addgene ID#12260), and envelope plasmid (VSV-G, Addgene ID#8454) were transfected into 293 T cells using X-treme GENE9 DNA transfection reagent (Roche) according to manufacturer's instructions. After 48 hr of transfection, virus containing supernatants from two 10 cm plates were collected and filtered through 0.45 μm membrane. Virus particles were concentrated by ultracentrifugation (1.5 hr at 80,000 g, 4°C) and resuspended in 200

µl of fresh DMEM (virus soup). For cell conversion, we applied 15 µl of virus soup per well (six-well plates). Materials and reagents are available upon request.

## Generation of iMNs

Human fibroblasts were seeded at 0.3–1 × 10⁴ cells on gelatin-coated 6-well plates. On the next day, the fibroblasts were infected with 2 ml of 15 µl virus soup (lentiviral vector carrying human *OCT4*) and fibroblast medium mixture containing 6 µg/ml protamine sulfate. The medium was replaced with fresh fibroblast medium after 24 hr of infection. At 3 days post-infection, the medium was switched to neural induction medium (DMEM/F12 (GIBCO) supplemented with N2 (GIBCO), penicillin/streptomycin, 10 ng/ml bFGF (Peprotech), and 10 ng/ml epidermal growth factor (EGF, Peprotech) and 10 ng/ml laminin). Morphologically changed infected cells (compact colonies of neural progenitor-like cells) were mechanically isolated by a glass micropipette and transferred into new wells individually. The cells were expanded in neural induction medium and iMNICs were established after 1 or 2 passages. For further MN induction, we used two methods, sphere culture and adherent culture.

[Sphere culture method] iMNICs were seeded at 5 × 10⁴ cells on gelatin-coated six-well in neural induction medium. On the next day, seeded iMNICs were infected with 2 ml of 15 µl virus soup (lentiviral vector carrying human *LHX3*) and neural induction medium mixture containing 6 µg/ml protamine sulfate. After 24 hr, the medium was replaced with defined MN induction medium (DMEM/F12 and neurobasal medium (GIBCO) supplemented with N2 (GIBCO), B27 (GIBCO), penicillin/streptomycin, 50 ng/ml SHH (Peprotech), and 10 ng/ml IGF-1). By 5–7 days, the cells formed clusters and MN spheres appeared. The floating MN spheres were re-plated on PDL/laminin-coated plate in MN maturation medium (DMEM/F12 and neurobasal medium (GIBCO) supplemented with N2 (GIBCO), B27 (GIBCO), penicillin/streptomycin, 50 ng/ml SHH (Peprotech), 10 ng/ml IGF-1, 10 ng/ml BDNF, 10 ng/ml GDNF, 10 ng/ml CNTF and 10 ng/ml NT3). MN-like cells outgrew gradually and further matured for additional 7–14 days.

[Adherent culture method] iMNICs were seeded at 5 × 10⁴ cells on PDL/laminin-coated plate in neural induction medium. On the next day, seeded iMNICs were infected with 2 ml of 15 µl virus soup (lentiviral vector carrying human *LHX3*) and neural induction medium mixture containing 6 µg/ml protamine sulfate. After 24 hr, the medium was switched into MN induction medium for 7 days. In this process, the MN spheres did not appear. We switched the medium into MN maturation medium and matured for additional 7–14 days.

## Immunocytochemistry (ICC)

Immunostaining was performed as previously described (*Kim et al., 2015*). The primary antibodies used for ICC are listed in *Supplementary file 1*. The secondary antibodies were diluted in PBS and applied for 1 hr: Alexa Fluor 488/555/568/594 anti-mouse IgG, IgG1, IgM, anti-chicken IgY, anti-rabbit IgG, and anti-goat IgG (Invitrogen, 1:1,000). Nuclei were stained with DAPI (Invitrogen).

## C2C12 myotube co-culture

C2C12 myoblasts (ATCC) were expanded in DMEM with 10% FBS and penicillin/streptomycin. When the culture reached 70% confluency, the medium was switched to 2% horse serum containing medium to induce multinucleated myotubes. iMNs were added to the myotubes in motor neuron medium to induce formation of neuromuscular junctions and spontaneous contractions. After 3–4 weeks, the myotube contractions were observed under the microscope and were inhibited by adding 100 µM curare (Sigma). Neuromuscular junctions were observed by labeling with α-bungarotoxin conjugated with Alexa 555 (Invitrogen, 1:200) and immunostaining with myosin heavy chain (MHC, DSHB), acetylcholine receptor (AChR, DSHB), and synaptic vesicle 2 (SV2, DSHB).

## RT–PCR and quantitative RT–PCR

DNA-free total RNA of iMNs was extracted using the RNeasy mini kit (Qiagen). Total RNA (500 ng) was used to synthesize cDNAs using SuperScript III reverse transcriptase (Invitrogen). RT–PCR was performed using recombinant Taq DNA polymerase (Invitrogen). qRT–PCR analysis was conducted on a LightCycler 480 instrument with SYBR Green I Master mix (Roche). The experiments were performed in triplicate, and expression was normalized to the housekeeping gene GAPDH. Gene

expression was measured by calculating Ct values. All of the experiments were conducted according to the manufacturer's instructions. The sequences of the primers used are listed in *Supplementary file 2*.

## Electrophysiology

Motor neurons induced from human fibroblasts were placed in a recording chamber and recognized visually by IR-DIC optics. Induced motor neurons were continuously perfused with artificial cerebrospinal fluid (ACSF) containing 125 mM NaCl, 2.5 mM KCl, 1.25 mM NaH2PO4, 25 mM NaHCO3, 15 mM glucose, 2 mM CaCl2, and 1 mM MgCl2 oxygenated with 95% $O_2$% and 5% $CO_2$ at 30–32℃. Borosilicate glass pipettes (2.5–3.5 MΩ) were pulled (P-1000, Sutter Instrument) and filled with potassium-based internal solution containing 133 mM KMeSO3, 3 mM KCl, 10 mM HEPES, 1 mM EGTA, 0.1 mM CaCl2, 8 mM Na2-phosphocreatine, 4 mM Mg-ATP, 0.3 mM Na3-GTP (290–300 mOsm, pH 7.3 with KOH) to make whole-cell configuration. Whole-cell patch clamp recording was performed by using Multiclamp 700B (Molecular Devices) and recording signals were filtered at 2 kHz, digitized at 10 kHz (PCI-6221, National Instruments). Recording data were monitored, acquired by WinWCP (Strathclyde Electrophysiology Software) and further analyzed offline by Clampfit 10.0 (Molecular Devices) and Prism 7.0 (GraphPad). To evaluate passive membrane properties of iMNs, hyperpolarizing step current (50 pA increment, 1000 ms duration) was injected in current clamp mode. To check active membrane properties of iMNs and generate AP, depolarizing step current (50 pA increment, 1000 ms duration) was injected. Electrophysiological properties of AP (threshold, frequency, amplitude, half-width) were analyzed from AP firings induced by 50 pA current injection. To isolate sodium current, whole-cell recording was conducted under voltage clamp mode and voltage was stepped from a holding potential of −70 mV to test potentials from −70 to −25 mV in 5 mV increments (200 ms duration). After recording sodium currents, tetrodotoxin (1 μM, Tocris) was bath-applied and voltage steps were repeated to confirm TTX-sensitivity of sodium currents measured.

## Growth curve and mean doubling time

Motor neuron intermediate cells ($1 \times 10^4$ cells) at P2 and P13 were seeded onto 12-well plates and cultivated for 10 days. The cells were collected from triplicate wells and manually counted every 24 hr using a hemacytometer (Marienfeld). The average cell numbers on each day were plotted, and the mean doubling time (mDT) was calculated based on the growth curve.

## Establish of compression spinal cord injury model

All animal experimental procedures were approved by Institutional Animal Care and Use Committee of Yonsei University College of Medicine (Seoul, Korea). Adult male Sprague-Dawley rats (Orient, Seongnam, Korea), weighing 200–220 g were used. Rats were housed in individual cages under standard laboratory conditions of 24–3C and 40–60% humidity, with 12 hr light–12 hr dark cycles, and enrichment conditions. All animals anesthetized with ketamine (100 mg/kg; Yuhan, Korea), xylazine (10 mg/kg; Bayer korea, Korea) and isotropy 100 (Troikaa Pharmaceuticals Ltd, India). A laminectomy was performed to expose the spinal cord, and the spinal cord was compressed at thoracic level 9 using self-closing forceps (Fine Science Tools, Canada) for 40 s. Following the injury, the muscle and skin were sutured with 3–0 Vicryl (Johnson and Johnson, Peterborough, Canada). Body temperature was maintained constant at 37C with a heating pad during surgery and the recovery period. After the procedure, cefazolin (25 mg/kg; Chong Kun Dang, Korea) was injected for 5 days. For immune suppression, Cyclosporine A (10 mg/kg, Chong Kun Dang, Korea) was administered to all animals until sacrifice. Also, animals were taken care of bladder system for urination.

Transplantation of iMNs were transplanted at 7 days after SCI. The animals were randomly divided into 3 groups: (1) control group that injected PBS (5 μl); (2) iMN spheres $1 \times 10^6$/5 μl; (3) HB9:GFP+ iMNs $1 \times 10^6$ /5μl. The pervious wound was reopened and cell injected by 27 gauge cannula connected to Hamilton syringe.

## Behavior test

All animals underwent behavioral analysis every week for 8 weeks after SCI to measure locomotor recovery. The Basso, Beattie, and Bresnahan (BBB) motor score was used to evaluate the quality of

hind limb movement during open field locomotion. In the first recovery phase, the range of joint movement and the presence of the foot closure on the floor were checked. In the second phase, recovery of weighted stepping was observed. In the third phase, gait coordination and tail movement were observed.

## Teratoma formation assay

All mice were purchased from Hyochang Science (Daegu, Korea). Animal handling was in accordance with animal protection guideline of Ulsan National Institute of Science and Technology (Ulsan, Korea). Teratoma formation assay was performed by subcutaneously injecting iMNICs (n = 4) and iMNs (n = 4) respectively on dorsal flank of athymic nude mice ($1 \times 10^6$ cells/mouse). After 12 months after injection, mice were sacrificed for analysis of teratoma formation.

## Isolation of iMNs

To isolate pure iMNs for analysis, we designed the plasmid containing puromycin resistance gene under expression of HB9 promoter. After transfection, cells expressing HB9 were selected by culturing in the presence of 0.5 µg/ml of puromycin for 5 days.

## Histology and immunohistochemistry (IHC)

For histological analysis, the rats were deeply anesthetized and perfused with PBS followed by 4% paraformaldehyde as well. The spinal cords were isolated and post-fixed in 4% paraformaldehyde overnight, and then immersed in 30% sucrose for 3 days. The tissues were embedded in frozen section compound (Leica) and sectioned at 16 µm in sagittal plane by a cryostat. IHC was performed as previously described (*Kim et al., 2015*). The primary antibodies used for IHC are listed in *Supplementary file 1*.

For hematoxylin and eosin (H and E) staining, the spinal cords were embedded in paraffin blocks. The paraffin blocks were sectioned at 4 µm in sagittal plane. Sections were immersed in Harris hematoxylin solution (Sigma) for 2 min to stain nucleus. Slides were then immersed briefly in 1% acid alcohol (1% HCl in 70% ethanol) and blued in 0.2% ammonium hydroxide, followed by staining with eosin Y solution (Sigma) for 30 s. Each step was followed by several washings with distilled water. The slides were dehydrated with ethanol, cleared with xylene, and mounted with mounting solution (Leica).

## Statistical analysis

All data in this article are presented as the means ± SD (standard deviation). Data from at least three independent samples were used for statistical analysis. ANOVA with post hoc testing was performed to compare BBB score. A p-value less than 0.05 was considered statistically significant. Statistical analysis was carried out using EXCEL, and SigmaPlot software.

## RNA-seq preparation and data analysis

Total RNAs were extracted from cells using the RNeasy mini kit (Qiagen) according to manufacturer's instructions. The quality of RNA was examined using Agilent 2100 Bioanalyzer. RNA integrity number (RIN) of all samples were higher than 8. Library sequencing was carried out on NovaSeq 6000 instrument. We generated 100 bp paired-end reads, with each library sequenced up to depth of 40 million fragments. We used HISAT2 (*Pertea et al., 2016*) to align the RNA-seq reads to the human reference genome GRCh38, and Cufflinks (*Trapnell et al., 2012*) to annotate them. We calculated the counts of aligned reads to each gene with HTSeq (*Anders et al., 2015*). We equalized the data and stabilized them through the log2 transform of the data plus one. RNA-seq data sequence summary is provided in *Supplementary file 5*, and related codes are provided in *Supplementary files 6* and *7*.

## RNA-seq data integration analysis

To complement the RNA-seq data generated in this work, we collected RNA-seq data from the Sequence Read Archive (SRA) database for wild-type motor neurons MN, wtMN-2 (SRR2038215) (*Ng et al., 2015*), wtMN-1 (SRR606336) (*Amoroso et al., 2013*) and human ESC H9 (SRR3647179) (*Kumamaru et al., 2018*). As with our own samples, we used HISAT2 to align the RNA-seq reads to

the human reference genome GRCh38, Cufflinks to annotate the mapped reads, and HTSeq to calculate the counts. We equalized the data and stabilized them through the log2 transform of the data plus one. To reduce the batch effect of the data integration, we used ComBat (*Johnson et al., 2007*). We used in-house software to merge the expression results into a single text file used in the downstream analysis in Matlab (MathWorks).

### Transcriptomics global analysis

The heatmap of the most highly variable transcripts, the hierarchical clustering dendrograms (calculated using the unweighted pair group method with arithmetic mean and Euclidean distance measure), and the Principal Component Analysis (PCA) were performed using in-house functions developed in Matlab (MathWorks).

### Gene-coverage count track plots

We sorted the alignment bam files with samtools (*Li et al., 2009*) and produced the bed files with bedtools (*Quinlan and Hall, 2010*). We developed a function in Matlab (MathWorks) that for each gene of interest takes the exon boundary information from the basic annotation file in gtf format from Gencode (https://www.gencodegenes.org/human/) version 33, and plots the gene-coverage count track plots, preserving the same scale for the tracks of the same gene in all the samples.

### Data availability

The data discussed in this publication have been deposited in NCBI's Gene Expression Omnibus (*Edgar et al., 2002*) and are accessible through GEO Series accession number GSE149664 .

## Acknowledgements

This work (S2566811) was supported by Tech Incubator Program for Startup (TIPS) funded Korea Ministry of SMEs and Startups. J-IK was supported by Basic Science Research Program through the National Research Foundation of Korea (NRF) funded by the Ministry of Science and ICT (2017R1C1B3005476). DG and MJA-B was supported by Grants of Instituto de Salud Carlos III (AC17/00012).

## Additional information

### Funding

| Funder | Grant reference number | Author |
| --- | --- | --- |
| Korea Ministry of SMEs and Startups | S2566811 | Jeong Beom Kim |
| National Research Foundation of Korea | 2017R1C1B3005476 | Jae-Ick Kim |
| Instituto de Salud Carlos III | AC17/00012 | Daniela Gerovska<br>Marcos J Araúzo-Bravo |
| European Union Eracosysmed/ H2020 | 643271 | Daniela Gerovska<br>Marcos J Araúzo-Bravo |
| Ministry of Economy and Competitiveness of Spain MINECO | BFU2016-77987-P | Daniela Gerovska<br>Marcos J Araúzo-Bravo |

The funders had no role in study design, data collection and interpretation, or the decision to submit the work for publication.

### Author contributions

Hyunah Lee, Conceptualization, Resources, Data curation, Software, Formal analysis, Validation, Investigation, Visualization, Methodology, Project administration; Hye Yeong Lee, Resources, Validation, Investigation, Methodology; Byeong Eun Lee, Investigation, Methodology; Daniela Gerovska, Marcos J Araúzo-Bravo, Validation, Investigation; Soo Yong Park, Methodology; Holm Zaehres,

Resources, Methodology; Jae-Ick Kim, Funding acquisition, Investigation, Methodology; Yoon Ha, Resources, Methodology, Project administration; Hans R Schöler, Resources, Supervision; Jeong Beom Kim, Conceptualization, Resources, Data curation, Supervision, Funding acquisition, Validation, Investigation, Methodology, Project administration

### Author ORCIDs
Hyunah Lee (iD) https://orcid.org/0000-0002-5560-5969
Daniela Gerovska (iD) https://orcid.org/0000-0003-0671-4277
Marcos J Araúzo-Bravo (iD) https://orcid.org/0000-0002-3264-464X
Jeong Beom Kim (iD) https://orcid.org/0000-0001-6230-8826

### Ethics
Human subjects: The experiments were carried out in accordance with documented standards of the Institutional Review Board of Ulsan National Institute of Science and Technology (UNIST) (UNISTIRB-15-17-C) for human cell experiments.

Animal experimentation: All animal experimental and surgical procedures on animals were performed in accordance with institutional protocols approved by the Institutional Animal Care and Use Committee of Yonsei University College of Medicine (2015-0327) for rat experiments, and the Institutional Animal Care and Use Committee of Ulsan National Institute of Science and Technology (UNIST) (UNISTIACUC-17-34) for mouse experiments.

### Decision letter and Author response
Decision letter https://doi.org/10.7554/eLife.52069.sa1
Author response https://doi.org/10.7554/eLife.52069.sa2

## Additional files

### Supplementary files
- Supplementary file 1. Primary antibodies used for ICC and IHC.
- Supplementary file 2. Primers used for quantitative RT-PCT and genomic PCR.
- Supplementary file 3. Summary of iMNIC induction from human fibroblast lines.
- Supplementary file 4. Characterization of established clones.
- Supplementary file 5. RNA seq data sequence summary.
- Supplementary file 6. Code for alignment and obtained alignment rates.
- Supplementary file 7. Code for obtaining genes counts and obtained statistics.
- Transparent reporting form

### Data availability
The data discussed in this publication have been deposited in NCBI's Gene Expression Omnibus and are accessible through GEO Series accession number GSE149664 (https://www.ncbi.nlm.nih.gov/geo/query/acc.cgi?acc=GSE149664). Source data files have been provided for Figure 1, 2, and 5.

The following dataset was generated:

| Author(s) | Year | Dataset title | Dataset URL | Database and Identifier |
|---|---|---|---|---|
| Lee H, Lee HY, Lee BE, Zaehres H, Park S, Kim JI, Ha Y, Gerovska D, Arauzo-Bravo MJ, Schoeler HR, Kim JB | 2020 | Sequentially induced motor neurons from human fibroblasts promote locomotor recovery in rodent spinal cord injury model | https://www.ncbi.nlm.nih.gov/geo/query/acc.cgi?acc=GSE149664 | NCBI Gene Expression Omnibus, GSE149664 |

The following previously published datasets were used:

| Author(s) | Year | Dataset title | Dataset URL | Database and Identifier |
|---|---|---|---|---|
| Amoroso MW, Croft GF, Williams DJ, O'Keeffe S, Carrasco MA, Davis AR, Roybon L, Oakley DH, Maniatis T, Henderson CE, Wichterle H | 2013 | Accelerated high-yield generation of limb-innervating motor neurons from human stem cells | https://www.ncbi.nlm.nih.gov/geo/query/acc.cgi?acc=GSE41795 | NCBI Gene Expression Omnibus, GSE41795 |
| Kumamaru H, Kadoya K, Adler AF, Takashima Y | 2018 | Comparison of human brain and spinal cord neural stem cells (NSCs) | https://www.ncbi.nlm.nih.gov/geo/query/acc.cgi?acc=GSE83107 | NCBI Gene Expression Omnibus, GSE83107 |

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
