## [Decision Letter]

**Acceptance summary:**

This study reports a novel method for the direct transdifferentiation of human fibroblasts to motor neurons based on the sequential expression of two transcription factors, *OCT4* and *LHX3*. The two-step protocol induces cells to pass through a state were cells keep dividing in principle allowing to generate large quantities of cells for clinical studies. The generated induced motor neurons show functional improvements in a pre-clinical model for spinal cord injury and could thus help to develop cell based therapies for motor neuron diseases.

**Decision letter after peer review:**

Thank you for submitting your article "Generation of induced motor neurons(iMNs) from human fibroblasts facilitates locomotor recovery after spinal cord injury" for consideration by *eLife*. Your article has been reviewed by three peer reviewers, one of whom is a member of our Board of Reviewing Editors, and the evaluation has been overseen by Didier Stainier as the Senior Editor.

The reviewers have discussed the reviews with one another and the Reviewing Editor has drafted this decision to help you prepare a revised submission.

Summary:

Lee, Kim et al. report a novel protocol for the direct conversion of human fibroblasts to induced motor neurons (iMN) through the sequential lentiviral expression of *OCT4* and *LHX3*. A key advancements of this work is that cells pass through a proliferative intermediate (iMNICs), that is distinct from iNPCs as it doesn't express standard progenitor markers, where cells can be expanded before proceeding to terminal iMN differentiation. They show functional improvement in vivo using a rat spinal cord injury model. Our consensus opinion is that your work is of interest, well written and goes beyond previously published studies that reported the generation of induced motor neurons through direct cell fate conversions. Authors claim that iMNICs represent a novel state that hasn't been reported before. While we find the subject of the study interesting, relevant and in principle of sufficient merit for publication in *eLife*, we jointly opine that authors should demonstrate more clearly the robustness of their reprogramming methods and further characterise the interesting iMNIC state.

Essential revisions:

There are two major concerns that we feel are essential to be addressed in a revised version.

1) Authors should characterise iMNICs using genome wide expression analysis (i.e. RNA-seq) to compare these cells to NPCs/NSCs, starting fibroblasts and iMN – ideally in reference to in vivo MNs. What are these cells? Do they still express fibroblasts genes? Expression of HOX genes critical for hindbrain and cervical identities should highlight in these datasets in addition to neural progenitor markers.

2) Authors should clarify how many fibroblast lines were used in the study to successfully make iMNs. Gender and age of donor cells should be given. From each fibroblast lines how many clones were expanded as iMNICs and converted into iMNs? This information is critical to evaluate the robustness of their method. The above gene expression profiling should include data from more than one of the established iMNIC lines.

[Editors' note: further revisions were suggested prior to acceptance, as described below.]

Thank you for resubmitting your work entitled "Generation of induced motor neurons (iMNs) from human fibroblasts facilitates locomotor recovery in rodent spinal cord injury model" for further consideration by *eLife*. Your revised article has been evaluated by Didier Stainier as the Senior Editor and a Reviewing Editor.

The manuscript has been improved but there are some remaining issues that need to be addressed before acceptance, as outlined below.

We appreciate that you have addressed the two main concerns identified by the reviewers during the initial evaluation of the study. In the revised manuscript you show that the reprogramming strategy could be repeated for one additional cell line and performed a more careful characterisation of the proliferative progenitor cell state as well as of the final iMNs using genome wide expression analysis. The overexpression of *OCT4* in human fibroblasts directs cells towards a proliferative *ISL1*-positive motor neuron progenitor that is remarkably distinct from neural progenitors despite the use of a culture medium that otherwise supports a neural progenitor state. Upon the subsequent overexpression of *LHX3*, post-mitotic induced motor neurons emerge that resemble their endogenous counterparts and could lead to functional improvements in a pre-clinical model for spinal cord injury. If the method can be applied to other cell lines and tissue types, these findings will be a significant advance in the field in particular for the possibility to generate authentic patient-specific cells at scale.

There remain some grammar and typographical errors throughout the manuscript and authors should carefully revise and edit the manuscript before publication. To better adhere with journal guidelines you may like to consider the alternative title 'Sequentially induced motor neurons from human fibroblasts facilitate locomotor recovery in a rodent spinal cord injury model'. In the Abstract I suggest rephrasing to 'Our strategy enables the scalable production of pure iMNs because of the transient acquisition of a proliferative iMN-intermediate cell stage which is distinct from neural progenitors.'

---

## [Author Response]

Essential revisions:There are two major concerns that we feel are essential to be addressed in a revised version.1) Authors should characterise iMNICs using genome wide expression analysis (i.e. RNA-seq) to compare these cells to NPCs/NSCs, starting fibroblasts and iMN – ideally in reference to in vivo MNs. What are these cells? Do they still express fibroblasts genes? Expression of HOX genes critical for hindbrain and cervical identities should highlight in these datasets in addition to neural progenitor markers.

We thank the reviewer for raising this point and we fully agree that it is necessary to characterize iMNICs and iMNs by using genome wide analysis. To address the request, we now have performed RNA-sequencing to evaluate global gene expression profiles of iMNICs and iMNs with other types of cells including: (1) neural progenitor cells (NPCs) defined from our previous study (Kim et al., 2009); (2) motor neurons (MN) derived from pluripotent stem cells (Amoroso et al., 2013; Ng et al., 2015); (3) fetal spinal cord tissue (Kumamaru et al., 2018); (4) fibroblasts as the control of parental cells before induction. Since we do not have access to obtain human spinal cord tissue or wild-type motor neurons, we downloaded RNA-seq datasets from GEO accessions GSE41795, GSE69175 (wild-type motor neurons) GSE83107 (fetal spinal cord tissue). iMNICs and iMNs clustered closely with wild-type MNs but separated from parental fibroblasts and NPCs (Figure 5A-D). Notably, gene expression patterns of iMNICs and iMNs are very similar with wild-type MN especially in MN genes and HOX genes which are essential for determination of MN columnar identity within spinal cord (Figure 5E). HOX genes are reported to be expressed at brachial and cervical levels (*HOX4-8*), thoracic level (*HOX8* and *HOX9*) and lumbar region (*HOX10* and *HOX11*) (Philippidou and Dasen, 2013). iMNICs and iMNs expressed cervical-to-lumbar HOX genes, suggesting our cells acquired cervical-to-lumbar spinal cord identity. In contrast, iMNICs and iMNs did not express NPC marker genes (SOX2, PAX6, OLIG2, SOX1, *ASCL1*, *FABP7*, and *EMX2*), and fibroblasts genes (COL1A1, COL5A1, *SNAI1*, and *SNAI2*) were down-regulated in iMNICs and iMNs (Figure 5F). Additionally, we conducted qPCR analysis to re-confirm the expression of HOX genes in iMNICs and iMNs. The mRNA level of HOX genes (*HOXC4*, *HOXC5*, *HOXC6*, *HOXA7*, *HOXC8*, *HOXA9*, and *HOXC9*) gradually increased during neural induction and highly expressed in iMNICs and iMNs compared to parental fibroblasts (Figure 5G). These results confirm that our iMNICs and iMN possess spinal cord identity, but not NPC. The data are now added as a new Figure 5 and Figure 5—figure supplement 1, and modified our manuscript in light of these results adding new subsection “Spinal cord identity of iMNICs and iMNs” in the Results section. We also contextualized this in result and Discussion section in our manuscript.

2) Authors should clarify how many fibroblast lines were used in the study to successfully make iMNs. Gender and age of donor cells should be given. From each fibroblast lines how many clones were expanded as iMNICs and converted into iMNs? This information is critical to evaluate the robustness of their method. The above gene expression profiling should include data from more than one of the established iMNIC lines.

We agree that fibroblast lines used for reprogramming should be presented more clearly. We now added a new table in Supplementary file 3 showing detailed information of two human fibroblasts lines, HF1 and HF2.

We appreciate the reviewers’ comments and suggestions and thank for prompting us to evaluate the robustness of our method. All presented data regarding generation of iMNICs and iMNs in our original manuscript were based on a single fibroblast line, HF1. To further strengthen the reproducibility and robustness of our protocol, we applied same method to additional fibroblast line, HF2. Generation of iMNs was successfully confirmed in HF2 (Figure 4—figure supplement 1). We could obtain proliferating iMNIC clones from HF2 as well and we could induce mature iMNs. This data is added in a Figure 4—figure supplement 1. We now recapitulated our experiments in Supplementary file 3, showing the number of established iMNIC clones and conversion efficiency of each fibroblast lines into iMNICs, assessed by counting the number of colonies emerged from seeded cells (Supplementary file 3). We also added a new table showing characteristics of established iMNIC clones (Supplementary file 4). All established iMNIC clones possessed proliferation capacity until more than passage 10 (P10~) and expressed *ISL1* gene. We could obtain HB9+iMNs from all established clones. We have performed gene expression profiling analysis by RNA-seq in two iMNIC clones, HF1-iMNIC6 and HF2-iMNIC6, which are generated from each of two fibroblast lines, HF1 and HF2, respectively (Figure 5). We indicated which clones were used for RNA-SEQ experiment in Supplementary file 4.

[Editors' note: further revisions were suggested prior to acceptance, as described below.]

The manuscript has been improved but there are some remaining issues that need to be addressed before acceptance, as outlined below.We appreciate that you have addressed the two main concerns identified by the reviewers during the initial evaluation of the study. In the revised manuscript you show that the reprogramming strategy could be repeated for one additional cell line and performed a more careful characterisation of the proliferative progenitor cell state as well as of the final iMNs using genome wide expression analysis. The overexpression of OCT4 in human fibroblasts directs cells towards a proliferative ISL1-positive motor neuron progenitor that is remarkably distinct from neural progenitors despite the use of a culture medium that otherwise supports a neural progenitor state. Upon the subsequent overexpression of LHX3, post-mitotic induced motor neurons emerge that resemble their endogenous counterparts and could lead to functional improvements in a pre-clinical model for spinal cord injury. If the method can be applied to other cell lines and tissue types, these findings will be a significant advance in the field in particular for the possibility to generate authentic patient-specific cells at scale.There remain some grammar and typographical errors throughout the manuscript and authors should carefully revise and edit the manuscript before publication. To better adhere with journal guidelines you may like to consider the alternative title 'Sequentially induced motor neurons from human fibroblasts facilitate locomotor recovery in a rodent spinal cord injury model'. In the Abstract I suggest rephrasing to 'Our strategy enables the scalable production of pure iMNs because of the transient acquisition of a proliferative iMN-intermediate cell stage which is distinct from neural progenitors.'

We appreciate this comments, we now rephrased title and Abstract as suggested.